# BOOSTING META-TRAINING WITH BASE CLASS INFORMATION FOR FEW-SHOT LEARNING

## ABSTRACT

Few-shot learning aims to learn a classifier that could be adapted to recognize new classes unseen during training with limited labeled examples. Meta-learning has recently become the most important framework for few-shot learning. Its training framework is originally a task-level learning method, such as Model-Agnostic Meta-Learning (MAML) and Prototypical Networks. And a recently proposed training paradigm, Meta-Baseline, that consists of sequential pre-training and meta-training stages, gains state-of-the-art performance. However, Meta-Baseline is not an end-to-end method, which means the meta-training stage can only begin after the completion of pre-training, leading to longer training time. Moreover, the two training stages would adversely affect each other, resulting in a decline in the latter training periods that is even lower than that of Prototypical Networks. In this work, motivated by the optimization method of stochastic variance reduced gradient, we propose a new end-to-end training paradigm consisting of two alternate loops. In the outer loop, we calculate the cross entropy loss on the whole training set but only update the final linear layer; while in the inner loop, we utilize the original meta-learning training mode to calculate the loss and incorporate the outer loss gradient to guide the parameter update. This training paradigm not only converges quickly but also outperforms the baselines, indicating that information from the overall training set and the meta-learning training paradigm could mutually reinforce one another.

## 1 INTRODUCTION

Deep learning has achieved remarkable success in the fields of text, image, and audio classification in recent years. Training these deep learning models often requires huge amount of data. However, when the available data is limited, the models often perform poorly due to the overfitting issue. This leads to the few-shot learning problem Koch et al. (2015); Lake et al. (2011): how can we train a model to achieve good results using a small amount of data when faced with new tasks? Meta-learning Sung et al. (2018) has recently been the most common framework to address the few-shot learning problem.

Meta-learning is divided into two stages: meta-training and meta-testing. Data used in the two stages are not intersected. Meta-training samples tasks from the whole training set. There are a support set and a query set for each task. The support set contains $N$ classes each with $K$ samples, formulated as $N$-way $K$-shot problem, while the query set contains $N$ classes each with $Q$ samples. Meta-learning aims to train a robust model during the meta-training phase, which can effectively classify the query set with only $N \times K$ support samples in new tasks during meta-testing.

Primary meta-learning training methods roughly fall into two categories, *i.e.*, optimization-based and metric-based, both of which are versions without pre-training. A good initialization parameter is learned during the meta-training of optimization-based methods, which can be quickly adapted to new tasks in the meta-testing stage. The goal of metric-based methods is to learn a good representation embedding in the meta-training stage. In the meta-testing stage, the support set and query set are mapped to the same high-dimensional embedding space, and the classification is carried out by measuring the distance between the two sets.

Meta-learning with pre-trained models emerges in recent years, which includes two stages. In the pre-training stage, the whole training classes of data are used to train the feature extractor through

supervised or unsupervised methods. In the meta-learning stage, the original meta-learning training method (such as Prototypical Networks) is used to retrain the feature extractor. The representative work in this category is Meta-Baseline, which outperforms the primary meta-learning methods and achieves new state-of-the-art performance.

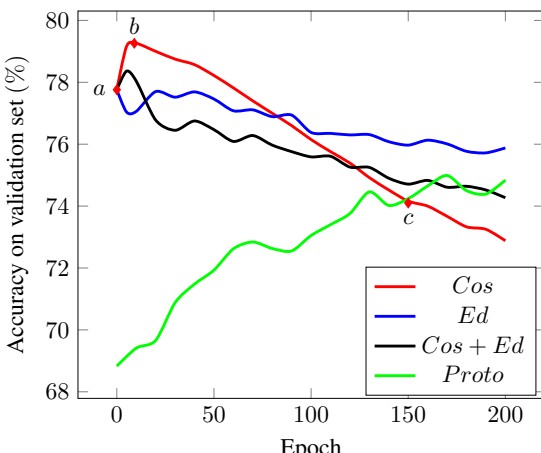

Figure 1: Accuracy of the second stage of Meta-Baseline with cosine similarity ($Cos$), Euclidean distance ($Ed$), cosine similarity plus Euclidean distance ($Cos + Ed$) on the validation dataset of $mini$ImageNet along with 5-way 5-shot training process. We also re-implement primary Protypical Networks ($Proto$) under the same setting.

However, as shown in Figure 1, we observe that in the second stage of Meta-Baseline, the performance on the validation set first increases, then decreases, and finally, is even lower than that of the Prototypical Networks, indicating that the training of the first and second stages have some negative impact on each other. Worse still, we could not mitigate the issue even if we change the metric, indicating the difficulty of utilizing the representation information of pre-training in the second stage of Meta-Baseline.

Besides, Meta-Baseline is not an end-to-end training method, which means the second training stage must wait until the first stage completes the convergence, resulting in a significantly longer overall training time. Moreover, the accuracy of the first and second stages is not proportional, indicating that the model with higher accuracy in the first stage may not perform better in the second stage. Therefore, multiple models must be trained in the second stage to achieve better results. In summary, simply concatenating the two stages of training is not helpful.

In this work, inspired by the optimization method of stochastic variance reduced gradient (SVRG) Johnson & Zhang (2013), we design a new end-to-end training approach to boost meta-training with base class information, termed Boost-MT . In this method, there are two loops that are executed alternately. In the outer loop, we calculate the classification loss of one large batch from the whole training set and only update the final linear layer. In the inner loop, we use the meta-learning method to calculate the loss of several divided episodic tasks and update the model by incorporating inner loss and outer loss. Our method not only can combine pre-training and meta-training into a single end-to-end model but also can quickly converge and achieve competitive results, which is more consistent with the deep learning paradigm. Additionally, our approach can avoid mutual subtraction between the two stages of Meta-Baseline.

In addition, our model is very simple and easy to use. There is almost no increase in the amount of parameters compared with the previous model. The only extra hyper-parameter is the proportion of outer and inner loops. We further conduct ablation experiments to explore the effectiveness of our model. Our main contributions are summarized as follows:

• We propose a new few-shot learning framework, that is end-to-end and can converge quickly, providing new motivations for future research on few-shot learning.

• To the best of our knowledge, this work is the first to adopt gradient information from base class to the training process of few-shot learning. In this way, the pre-training and meta-learning can guide each other, thus avoiding the mutual subtraction between the two stages observed in existing methods.

• We evaluate the proposed method on two benchmark datasets, $mini$ImageNet and $tiered$ImageNet datasets, and demonstrate its competitive performance compared to the state-of-the-art baselines.

## 2 RELATED WORK

This section reviews some popular methods on few-shot learning, that fall into three categories.

### 2.1 META-LEARNING FOR FEW-SHOT LEARNING

Most of the existing few-shot learning methods rely on the meta-learning framework, which can be roughly divided into two categories, *i.e.*, optimization-based Finn et al. (2017); Li et al. (2017); Nichol et al. (2018); Ravi & Larochelle (2017b); Raghu et al. (2020); Oh et al. (2021); Rusu et al. (2019) and metric-based Li et al. (2019); Oreshkin et al. (2018); Sung et al. (2018); Vinyals et al. (2016); Snell et al. (2017); Liu et al. (2019); Zhang et al. (2020).

The optimization-based methods aim to quickly learn the model parameters that can be optimized when encountering new tasks. The representative work of this category is MAML Finn et al. (2017), which obtains good initialization parameters through internal and external training in the meta-training stage. The model with good parameters can obtain good results with a few steps of update in the meta-testing stage. MAML has inspired many follow-up efforts, such as ANIL Raghu et al. (2020), BOIL Oh et al. (2021), and LEO Rusu et al. (2019).

Metric-based methods embed support images and query images into the same space and classify query images by calculating the distance or similarity. Prototypical Networks Snell et al. (2017) compute a prototype for each class and classify query images by calculating the Euclidean distance. Relation Networks Sung et al. (2018) calculate distances between support images and query images via a relation module. TADAM Oreshkin et al. (2018) boosts Prototypical Networks by metric scaling and metric task conditioning. DeepEMD Zhang et al. (2020) employs the Earth Mover's Distance (EMD) as a metric to compute a structural distance between dense support images and query images.

### 2.2 TRANSFER LEARNING FOR FEW-SHOT LEARNING

The transfer learning framework for few shot learning Gidaris & Komodakis (2018); Chen et al. (2019); Tian et al. (2020); Dhillon et al. (2020); Shen et al. (2021) uses a simple scheme to train a classification model on the overall training set. During testing, it removes the classification head and retains the feature extraction part, then trains a new classifier based on the support set from testing data. This approach yields results comparable to meta-learning.

Specifically, Dynamic Classifier Gidaris & Komodakis (2018) trains a feature extractor and a small sample category weight generator using all the base classes. In the testing stage, the weight generator can generate weight vectors for each level class with a few samples. Baseline++ Chen et al. (2019) replaces the linear classifier with a cosine classifier in the training stage, and directly trains the feature extractor and cosine classifier using all the base classes. During testing, it fixes the feature extractor and trains a new classifier with limited labeled examples in novel classes, and achieved good results. Good-Embed Tian et al. (2020) uses ordinary classification or self-supervised learning to train a feature extractor in the training stage. Then it fits a linear classifier during testing on features extracted by the pre-trained network for each task, and further applies self-distillation to the pre-trained network.

### 2.3 META-LEARNING WITH PRE-TRAINING

Inspired by the transfer learning for few-shot learning methods, several recent studies Chen et al. (2021); Xie et al. (2022); Hu et al. (2022); Yang et al. (2022); Ye et al. (2020); Wertheimer et al.

(2021) propose to use a meta-learning model with pre-training. Meta-Baseline Chen et al. (2021) is the first of this kind, which explores the advantages of the overall classification model and the meta-learning model, and proposes a baseline method that continues the meta-learning in the converged classifier by using the evaluation measure of cosine nearest centroid.

Many subsequent few-shot learning methods follow the paradigm of Meta-Baseline. For instance, Meta DeepBDC Xie et al. (2022), based on a typical transfer learning framework of Good-Embed, uses plenty of annotation data to train a better basic model that obtains the embedded features of the image and then uses the Brownian distance covariance measurement for meta-learning training based on the framework of Prototypical Networks. Contrastive-FSL Yang et al. (2022) proposes a novel contrastive learning-based framework that seamlessly integrates contrastive learning into both pre-training and meta-training. P>M>F Hu et al. (2022) uses an unsupervised training method to obtain feature extractors in pre-training, then uses meta-learning to optimize the model. It finally fine-tunes the model using sample of the support set in the novel class, pushing the application of this method to the limit.

Although the meta-learning method with pre-training has achieved good results, it is not an end-to-end method and does not conform to the deep learning paradigm. In this work, we propose a new end-to-end training framework with outer loop and inner loop, that leverages the information from the base class to guide meta-learning instead of relying on direct pre-training.

## 3 METHODOLOGY

We first establish preliminaries on few-shot learning in Section 3.1, then we briefly introduce the flow of the Meta-Baseline method in Section 3.2 and finally describe our method in Section 3.3.

### 3.1 NOTATION

Given a labeled dataset $\mathcal{D}^{base}$ with a large number of images, the goal of few-shot learning is to learn concepts in novel classes $\mathcal{D}^{novel}$. Note that $\mathcal{D}^{base}$ and $\mathcal{D}^{novel}$ are not intersected. In meta-training stage, the few-shot learning problem randomly sample many tasks $\mathcal{T} = \{(\mathcal{D}_t^{spt}, \mathcal{D}_t^{qry})\}_{t=1}^{T}$ from $\mathcal{D}^{base}$. The support set $\mathcal{D}_t^{spt} = \{(\mathbf{x}_i^t, y_i^t)\}_{i=1}^{NK}$ and the query set $\mathcal{D}_t^{qry} = \{(\mathbf{x}_j^t, y_j^t)\}_{j=1}^{NQ}$ in each task are sampled under the $N$-way $K$-shot setting (support set consists of $N$ classes each with $K$ images and query set consists of $N$ classes each with $Q$ images). $S_n$ denotes the set of examples labeled with class $n$ in each task.

### 3.2 META-BASELINE

Meta-Baseline includes two training stages. The first stage is for the pre-training. A classification model is trained on $\mathcal{D}^{base}$. The model is composed of feature extractor $f_\theta$ and classifier $fc_\omega$. The second stage is meta-learning training, which calculates the distance between features using metric learning. First, Meta-Baseline sends the support set into the feature extractor to obtain each sample feature and averages all sample features of each class to get the prototype of each class $n$:

$$\mathbf{c}_n = \frac{1}{|S_n|} \sum_{(\mathbf{x},y) \in S_n} f_\theta(\mathbf{x}). \tag{1}$$

Similarly, a query point $\mathbf{x}$ is sent into the feature extractor to obtain its feature. The cosine similarity between $\mathbf{x}$ and the support set prototypes is calculated to obtain the probability that $\mathbf{x}$ belongs to class $n$:

$$p_\theta(y = n|\mathbf{x}) = \frac{exp(d(f_\theta(\mathbf{x}), \mathbf{c}_n))}{\sum_{n=1}^{N} exp(d(f_\theta(\mathbf{x}), \mathbf{c}_n))}, \tag{2}$$

where $d(\cdot, \cdot)$ denotes the cosine similarity between two vectors. Finally, the cross entropy $\mathcal{L}$ is used as the loss function.

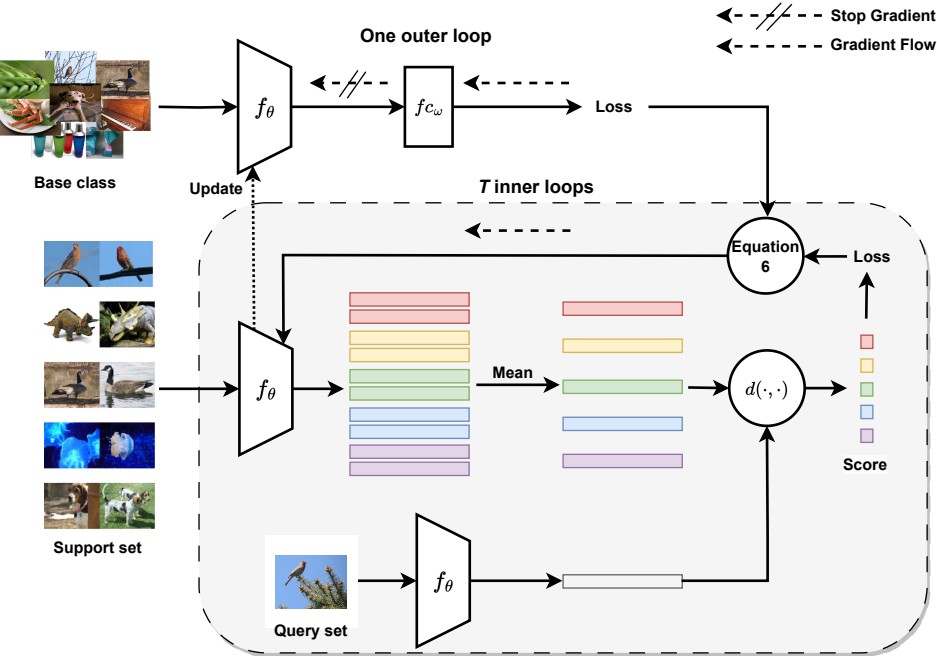

Figure 2: The main framework of our model. In the outer loop, we calculate the classification loss of one large batch from base classes and only update the linear classifier. In the inner loop, we use the meta-learning method to calculate the loss of tasks and update the model by inner loss and outer loss. The two loops are executed alternately, with $T$ inner loops per one outer loop.

### 3.3 BOOSTING META-TRAINING WITH BASE CLASS INFORMATION

As shown in Figure 2, our method consists of outer loop and inner loop. The outer loop is executed $S$ times in one epoch, and each outer loop corresponds to $T$ inner loops. The network model consists of a feature extractor $f_\theta$ used in both outer loop and inner loop, a classification header $fc_\omega$ used in outer loop, and a cosine similarity measurement function $d(\cdot, \cdot)$ used in inner loop. We randomly initialize $f_\theta$ and $fc_\omega$.

In each outer loop, we first randomly sample a large set $\mathcal{D}^{out}$ with $N_b$ images from $D^{base}$. Then we calculate the loss of $D^{out}$ by Equation 3 and only update $fc_\omega$. When all outer loops in one epoch are done, all samples in $\mathcal{D}^{base}$ are traversed once. Before the first inner loop begins, we randomly sample $T$ tasks $\mathcal{T} = \{(\mathcal{D}_t^{spt}, \mathcal{D}_t^{qry})\}_{t=1}^{T}$ from $\mathcal{D}^{base}$ as standard few-shot learning.

$$\tilde{\mu} = \frac{1}{N_b} \sum_{(\mathbf{x}, y) \in \mathcal{D}^{out}} \mathcal{L}(fc_{\tilde{\omega}_{s-1}}(f_{\tilde{\theta}_{s-1}}(\mathbf{x})), y), \tag{3}$$

In each inner loop, one task is calculated at a time. The feature extractor $f_\theta$ extracts the embedded representations of the support set and query set. We calculate the prototype of each class in the support set by Equation 1. For each sample in the query set, we use cosine similarity measurement $d(\cdot, \cdot)$ and predict its label by Equation 2. The loss of query point $\mathbf{x}$ is calculated by the cross entropy loss. The inner loss of the query set is calculated by $f_\theta$ with inner loop parameters and cosine similarity measurement $d(\cdot, \cdot)$, as in Equation 4. The outer loss of the query set is calculated by $f_\theta$ with outer loop parameters and $d(\cdot, \cdot)$, as in Equation 5. The inner and outer losses of the query set and the outer loss of $\mathcal{D}^{out}$ will all be used to update the parameters of $f_\theta$, as in Equation 6. In the next inner loop, the updated parameters in the previous inner loop are used to calculate inner-loss until all inner loops are done. When all the inner loops are done, the parameters of $f_\theta$ in the next outer loop are inherited from the previous inner loop. The details are shown in Appendix A.1, Algorithm 1.

$$\sigma = \frac{1}{NQ} \sum_{(\mathbf{x},y) \in \mathcal{D}_t^{qry}} \mathcal{L}(p_{\theta_{t-1}}(y = n|\mathbf{x}), y), \tag{4}$$

$$\tilde{\sigma} = \frac{1}{NQ} \sum_{(\mathbf{x},y) \in \mathcal{D}_t^{qry}} \mathcal{L}(p_{\tilde{\theta}_{s-1}}(y = n|\mathbf{x}), y) \tag{5}$$

$$\theta_t = \theta_{t-1} - \beta(\nabla\sigma - \nabla\tilde{\sigma} + \nabla\tilde{\mu}), \tag{6}$$

As analyzed in Section 1, a fixed embedding space from pre-training is difficult to exploit for meta-training. Unlike the meta-learning method with pre-training, although we calculate the loss on the base classes in the outer loop, we do not update parameters of the feature extractor, avoiding the direct update of representation information by $\mathcal{D}^{base}$. Moreover, our end-to-end method can still utilize the information from $\mathcal{D}^{base}$ by updating inner parameters through outer loss and inner loss, which avoids mutual subtraction between the two stages and boosts meta-training.

## 4 EXPERIMENTS

In this section, we describe the results of experiments, examining the properties of our model and comparing the performance of our method against various baselines on two common benchmark datasets. We also conduct experiments on cross-domain scenarios to show the transfer capability of our method. In the end, we do ablation study to show the effectiveness of our method and the parameter study to show the impact of the number of inner loops per outer loop.

### 4.1 DATASETS

In the standard few-shot classification task, commonly used datasets are Omniglot Lake et al. (2011), CUB-200-2011 (Birds) Wah et al. (2011), $mini$ImageNet Ravi & Larochelle (2017a), and $tiered$ImageNet Ren et al. (2018). Omniglot is used more often in early studies. Many later algorithms have achieved very high accuracy in training and testing on this dataset. The scale of CUB dataset is between $mini$ImageNet and $tiered$ImageNet, so we use $mini$ImageNet and $tiered$ImageNet for few-shot classification, which are also the two most commonly used datasets. We also conduct cross-domain experiments by training on $mini$ImageNet and testing on CUB.

The $mini$ImageNet dataset was created by the authors of Matching Network Vinyals et al. (2016) and is currently the most popular benchmark. It contains 100 categories sampled from ILSVRC-2012 Krizhevsky et al. (2012). Each category contains 600 images, and the size of each image is 84 × 84, with 60,000 images in total. Under standard settings, it is randomly partitioned into training, validation and testing sets, each containing 64, 16 and 20 categories.

The $tiered$ImageNet dataset is another commonly used benchmark. Although it is also from ILSVRC-2012, compared to the former, it has a more extensive data scale. There are 34 super-categories, divided into training, validation and testing sets, with 20, 6 and 8 super-categories, respectively. It contains 608 categories, corresponding to 351, 97 and 160 categories in each partitioned dataset. This dataset is more challenging because it is more difficult for the model to identify samples of different categories from the same super-categories. The base classes and novel classes come from different super-categories.

The CUB dataset contains 200 classes and 11,788 images in total. Following the protocol of Hilliard et al. (2018) Wah et al. (2011), we randomly split the dataset into 100 base, 50 validation, and 50 novel classes.

### 4.2 IMPLEMENTATION DETAILS

In order to keep consistent with Meta-Baseline, we use the ResNet-12 network backbone with a feature dimension of 512. For the outer loop, we use the SGD optimizer with momentum 0.9 to update the parameters in linear layer, the learning rate starts from 0.1, and the decay factor is set

Table 1: Comparison to prior works on $mini$ImageNet. Average 5-way accuracy with 95% confidence interval.

| Model | Backnone | 1-shot | 5-shot |
|---|---|---|---|
| Baseline++ Chen et al. (2019) | ResNet-18 | 51.87 ± 0.77 | 75.68 ± 0.63 |
| MataOptNet Lee et al. (2019) | ResNet-12 | 62.64 ± 0.61 | 78.63 ± 0.46 |
| Shot-Free Ravichandran et al. (2019) | ResNet-12 | 59.04 ± 0.43 | 77.64 ± 0.39 |
| TADAM Oreshkin et al. (2018) | ResNet-12 | 58.50 ± 0.30 | 76.70 ± 0.30 |
| MTL Sun et al. (2019) | ResNet-12 | 61.20 ± 1.80 | 75.50 ± 0.80 |
| SLA-AG Lee et al. (2020) | ResNet-12 | 62.93 ± 0.63 | 79.63 ± 0.47 |
| ProtoNets + TRAML Li et al. (2020) | ResNet-12 | 60.31 ± 0.48 | 77.94 ± 0.57 |
| Classifier-Baseline Chen et al. (2021) | ResNet-12 | 58.91 ± 0.23 | 77.76 ± 0.17 |
| ProtoNets Snell et al. (2017) | ResNet-12 | 60.37 ± 0.83 | 78.02 ± 0.57 |
| Meta-Baseline Chen et al. (2021) | ResNet-12 | 63.17 ± 0.23 | 79.26 ± 0.17 |
| Boost-MT (ours) | ResNet-12 | **64.01 ± 0.97** | **81.00 ± 0.59** |

Table 2: Comparison to prior works on $tiered$ImageNet. Average 5-way accuracy with 95% confidence interval.

| Model | Backnone | 1-shot | 5-shot |
|---|---|---|---|
| LEO Rusu et al. (2019) | WRN-28-10 | 66.33 ± 0.05 | 81.44 ± 0.09 |
| MetaOptNet Lee et al. (2019) | ResNet-12 | 65.99 ± 0.72 | 81.56 ± 0.53 |
| MTL Sun et al. (2019) | ResNet-12 | 65.62 ± 1.80 | 80.61 ± 0.90 |
| AM3 Xing et al. (2019) | ResNet-12 | 67.23 ± 0.34 | 78.95 ± 0.22 |
| Shot-Free Ravichandran et al. (2019) | ResNet-12 | 66.87 ± 0.43 | 82.64 ± 0.43 |
| DSN-MR Simon et al. (2020) | ResNet-12 | 67.39 ± 0.82 | 82.85 ± 0.56 |
| ProtoNets + Rotation Su et al. (2020) | ResNet-18 | – | 78.90 ± 0.70 |
| ProtoNets Snell et al. (2017) | ResNet-12 | 65.65 ± 0.92 | 83.40 ± 0.65 |
| Classifier-Baseline Chen et al. (2021) | ResNet-12 | 68.07 ± 0.26 | 83.74 ± 0.18 |
| Meta-Baseline Chen et al. (2021) | ResNet-12 | 68.62 ± 0.27 | 83.74 ± 0.18 |
| Boost-MT (ours) | ResNet-12 | **69.73 ± 0.71** | **84.91 ± 0.49** |

to 0.1. For the inner loop, we set 10 tasks to update the feature extractor. On $mini$ImageNet, we train 100 epochs with batch size 128, and the learning rate decays at epoch 30 and 60. On $tiered$ImageNet, we train 100 epochs with batch size 256, and the learning rate decays at epoch 30 and 60. We also use commonly standard data augmentation methods, including random resized loop and horizontal flip. However, we do not set the cosine scaling parameter $\tau$. During the meta-testing, we randomly sample 1500 tasks with 15 query images per class and report the mean accuracy together with the corresponding 95% confidence interval.

## 4.3 RESULTS ON STANDARD BENCHMARKS

Following the standard setting of few-shot learning, we conduct experiments on $mini$ImageNet and $tiered$ImageNet, and the results are shown in Tables 1 and 2, respectively. As in the previous work, we use the validation set to select models. On the two datasets, we can observe that the proposed method of Boost-MT consistently achieves competitive results compared to the state-of-the-art Meta-Baseline method on both the 5-way 1-shot and 5-way 5-shot tasks. Among them, for the $mini$ImageNet dataset, we outperform Meta-Baseline by 0.94% and 1.74% on 5-way 1-shot and 5-way 5-shot, respectively; for the $tiered$ImageNet dataset, we outperform Meta-Baseline by 1.11% and 1.17% on 5-way 1-shot and 5-way 5-shot, respectively. As shown in Table 2, the second stage of Meta-Baseline can hardly gain improvement from the pre-training stage on $tiered$ImageNet, indicating that Meta-Baseline does not play the role of meta-learning. Our method outperforms ProtoNets, Classifier-Baseline and Meta-Baseline, showing that we make good use of both base classes information and meta-learning. It is worth mentioning that, in order to make a fair comparison with Meta-Baseline, we only change the training methods according to their settings and also do not use other more sophisticated structural designs or modules.

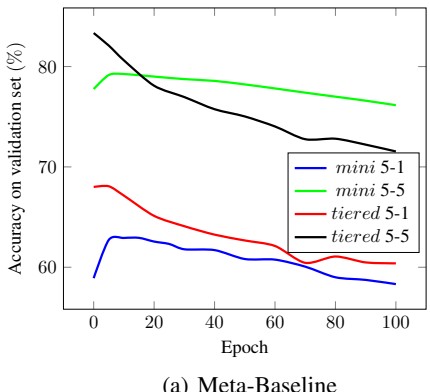

(a) Meta-Baseline

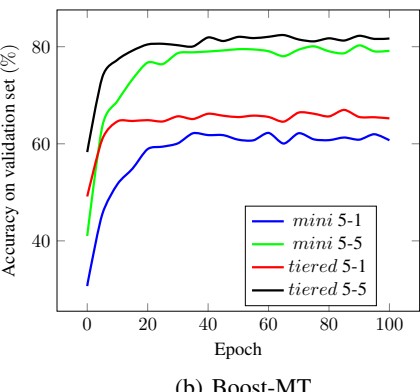

(b) Boost-MT

Figure 3: Accuracy of the validation set during training. (a) Accuracy of the second stage of Meta-Baseline on the validation set of $mini$ImageNet and $tiered$ImageNet along with the training process; and (b) accuracy of our method Boost-MT on the validation set of $mini$ImageNet and $tiered$ImageNet along with the training process. 5-1 denotes the 5-way 1-shot problem and 5-5 denotes the 5-way 5-shot problem.

In addition, when the outer loss is directly used to update the model parameters and remove the inner loop, our method will degenerate to the pre-training method Chen et al. (2019). When removing the outer loop, our method is a meta-learning method similar to Prototypical Networks Snell et al. (2017). From the experimental results, we can see that our results are superior to the two baseline methods.

## 4.4 COMPARISON WITH META-BASELINE

We reproduce Meta-Baseline and draw the performance on $mini$ImageNet and $tiered$ImageNet validation classes during the meta-learning stage. As shown in Figure 3(a), although the meta-learning stage improves the model performance on the validation set during the first few epochs, the performance of the model continues to decline after that.

We also draw the performance on validation classes of our model during training. As shown in Figure 3(b), our model achieves good performance and converges quickly. Our model begins to converge at the $40th$ epoch on $mini$ImageNet and keeps relatively stable fluctuations since then. Also, on the $tiered$ImageNet dataset, the model begins to converge at the $35th$ epoch. Compared with Meta-Baseline and many other few-shot learning methods Finn et al. (2017); Raghu et al. (2020); Oh et al. (2021), the convergence is faster and more stable, indicating that our model is more consistent with the paradigm of deep learning.

We further evaluate our model in cross-domain scenarios. We train our model on $mini$ImageNet and then test on CUB. The results in Table 3 show that our model outperforms Meta-Baseline by 1%, indicating that our model has stronger domain transfer capability.

## 4.5 FURTHER ANALYSIS

In this subsection, we perform ablation experiments to analyze how each component affects the performance of our method and do comparison under 5-way setting on $mini$ImageNet using the ResNet-12 backbone. We mainly study the following two components.

In order to verify the effectiveness of our optimization method, we design a comparative method called Vanilla. In this method, the loss of outer loop is not used to guide the parameter update of the inner loop but update the parameters of the feature extractor and linear layer. Then the parameters of the feature extractor are further updated by calculating the loss of query set in the inner loop. After the inner loop tasks are completed, we re-enter the outer loop training again, and execute alternately until the end. As shown in Table 4, it can be found that the improvement of the latter is significant.

Table 3: Comparison with Meta-Baseline for 5-way classification in cross-domain scenarios. †: reproduced using our setting.

| Model | Backnone | 1-shot | 5-shot |
|---|---|---|---|
| Meta-Baseline[†] | ResNet-12 | $44.69 \pm 0.59$ | $61.60 \pm 0.54$ |
| Boost-MT (ours) | ResNet-12 | $45.18 \pm 0.60$ | $62.91 \pm 0.57$ |

Table 4: Comparison to method without outer gradient guidance on $mini$ImageNet. We show the average 5-way accuracy with 95% confidence interval.

| Model | Backnone | 1-shot | 5-shot |
|---|---|---|---|
| Vanilla | ResNet-12 | $62.88 \pm 0.67$ | $76.19 \pm 0.50$ |
| Boost-MT (ours) | ResNet-12 | $64.01 \pm 0.97$ | $81.00 \pm 0.59$ |

Table 5: The average 5-way 5-shot performance of our model under different number of inner loops per outer loop on $mini$ImageNet.

| $T$ | 1 | 5 | 10 | 15 | 20 |
|---|---|---|---|---|---|
| Accuracy | 76.41 | 77.64 | 79.66 | 78.57 | 77.72 |

This further shows that using gradient information from the base class is more beneficial to the meta-training of the model than the parameter weight.

We further study the impact of different numbers of inner loops per outer loop. The results are shown in Table 5.When the number is either too small or too big, the performance of the model will decline. The performance is good when the number of inner loops is between 5 and 15. In our final experiment, the number of inner loops is set to 10. In the outer loop, the feature extractor and classifier are utilized to calculate the classification loss to achieve class transferability. In the inner loop, the feature extractor and metric module work together to calculate the loss for each specific few-shot classification task. This process allows the model to adapt and fine-tune its parameters specifically for the given few-shot classification task. We speculate that the number of inner loops mainly affects the update of internal parameters. When the number of internal tasks is too small, the number of parameter iterations is less, and the meta-training is not sufficient. When the number of inner loops is too large, the information of the base class could not be used effectively.

## 5 CONCLUSION

In this work, we propose a new few-shot learning framework Boost-MT, which boosts meta-training with base class information. Our experiments indicate that simply concatenating pre-training and meta-training like Meta-Baseline can not make full use of base class information because it is difficult for meta-training to measure the fixed representation from pre-training. As an end-to-end framework, our method leverages the gradient information from the base class to guide meta-learning, avoid mutual subtraction of pre-training and meta-training, converge quickly, and outperform the Meta-Baseline. To explore the relationship between pre-training and meta-training, we further conduct an experiment where we remove the outer loop loss from inner model update, and the model in both loops was trained by stochastic gradient descent. That is, the outer loop directly updates the model parameters and passes to the inner loop to continue meta-training. The performance of the derived method is poor, demonstrating that the loss of pre-training is instructive to the process of meta-training, and it is unwise to simply combine the two stages or consider them separately. The comprehensive experimental results show that our proposed method achieves competitive performance on two well-known benchmarks $mini$ImageNet and $tiered$ImageNet.

We will investigate possible improvement on Boost-MT, as a new framework for meta-training, in our future work. Also, our experiments have demonstrated that the base class information is useful for meta-learning, that also deserves further exploration in future works.

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

# A APPENDIX

## A.1 ALGORITHM 1

We present the full algorithm for our method in Algorithm 1.In this method, there are two loops that are executed alternately. In the outer loop, we calculate the classification loss of one large batch from the whole training set and only update the final linear layer. In the inner loop, we use the meta-learning method to calculate the loss of several divided episodic tasks and update the model by incorporating inner loss and outer loss.

---

**Algorithm 1** The Boost-MT Method

---

1: **Input:** base classes $\mathcal{D}^{base}$, feature extractor with parameters $\theta$: $f_\theta$, classification header with parameters $\omega$: $fc_\omega$, hyperparameters on step size: $\alpha, \beta$
2: **Output:** $\tilde{\theta}_S$
3: **Require:** RandSample($\mathcal{D}, N$): randomly sample $N$ images from $\mathcal{D}$
4: Initialize $\tilde{\theta}_0, \tilde{\omega}_0$
5: **for** $s = 1, 2, 3, \ldots, S$ **do**
6:     $\mathcal{D}^{out} \leftarrow$ RandSample($\mathcal{D}^{base}, N_b$)
7:     Compute the loss of $\mathcal{D}^{out}$ by Equation 3
8:     $\tilde{w}_s = \tilde{w}_{s-1} - \alpha \nabla \tilde{\mu}$
9:     $\theta_0 = \tilde{\theta}_{s-1}$
10:     Sample $\mathcal{T} = \{(\mathcal{D}_t^{spt}, \mathcal{D}_t^{qry})\}_{t=1}^T$ from $\mathcal{D}^{base}$
11:     **for** $t = 1, 2, 3, \ldots, T$ **do**
12:         Compute the prototype of the $t$ task by Equation 1
13:         Compute the probability of the query point **x** by Equation 2
14:         Compute the inner loss of $\mathcal{D}_t^{qry}$ with inner loop parameters by Equation 4
15:         Compute the outer loss of $\mathcal{D}_t^{qry}$ with outer loop parameters by Equation 5
16:         Update $\theta$ based on Equation 6
17:     $\tilde{\theta}_s = \theta_T$
18: **return** $\tilde{\theta}_S$

---

