# OpenReview forum: "Boosting Meta-Training with Base Class Information for Few-Shot Learning"
_ICLR.cc/2024/Conference — ICLR 2024 Conference Withdrawn Submission_

### Official Review · Reviewer_6HPg · 2023-10-31

**Soundness:** 2 fair
**Presentation:** 2 fair
**Contribution:** 1 poor
**Rating:** 3
**Confidence:** 4

**Summary:**

The paper tackles the problem of few-shot learning with meta-learning. The authors propose an improvement to the Meta-Baseline's two-stage training procedure, that would allow to train the network in one stage, end-to-end, which is potentially faster and simpler.

**Strengths:**

The authors demonstrate encouraging performance on the tired-imagenet and mini-imagenet.

**Weaknesses:**

- One of my concerns is the motivation of the paper given in the introduction. Why is it important to have end-to-end training and why are we not happy with the two-stage Meta-Baseline training? I think 2-stage training makes a lot of sense as one can simply take any pre-trained model. It is known that just throwing a backbone pre-trained on more data benefits few-shot learning. Having to re-train this backbone for few-shot learning specifically would be unnecessary use of time and resources.
- The authors comment on figure 1 “as shown in Figure 1, we observe that in the second stage of Meta-Baseline, the perfor- mance on the validation set first increases, then decreases, and finally, is even lower than that of the Prototypical Networks, indicating that the training of the first and second stages have some negative impact on each other”. Is in it just called overfitting? The network has already been trained on the same dataset with a slightly different objective. It is no surprise that fine-tuning further will lead to overfitting faster.
- The central result of the paper -  Equation 6 - is given without explanations and it is hard to understand why it is valid.
While the authors suggest that their method is more efficient in terms of wall-clock time, there is no justification. To support this claim, it would be nice to see how much time it takes to train the Meta-Baseline, i.e., the classification pre-training + several epochs of meta-finetuning, vs full training of the proposed method.
- It is nice to see encouraging results on the mini-ImageNet and tired-ImageNet dataserts, however these are old and non-informative datasets at this point. It would be nice to see some results on meta-dataset [A].
- Last but not least, the authors suggest that one-stage training would be conceptually simpler, however after reading this paper I have the opposite feeling. Training Meta-Baseline, even though in two-stages, is a streamlined process with simple classification pre-training and simple meta-learning fine-tuning. On the other hand, in the proposed paper, it seems quite cumbersome to desing the inner and outer loop updates, and tune their hyperparameters, such as the number of iterations in each loop.


[A] Triantafillou at al, Meta-dataset: A dataset of datasets for learning to learn from few examples, 2019

**Questions:**

- Could the authors give more explanations? I could not find sufficient details in the paper.

**Details Of Ethics Concerns:**

No concerns

---

### Official Review · Reviewer_hGNt · 2023-11-01

**Soundness:** 3 good
**Presentation:** 3 good
**Contribution:** 3 good
**Rating:** 6
**Confidence:** 3

**Summary:**

The paper proposed an updated method based on the Meta-Baseline few-shot image classifier that combines simultaneous outer updates from base class classification and inner updates from episodic training to learn an improved few-shot classifier. The method named Boost-MT first initializes a feature extractor and classifier and then during the outer-loop makes updates only to the classifier based on the cross entropy loss. Then, for T times, the inner loop is completed where the model is trained episodically but the propagated loss is a convex combination of the inner and outer losses. Experiments on CUB, mini- and tiered-ImageNet benchmarks show competitive performance with notable improvements over Meta-Baseline.

**Strengths:**

- The proposed method provides a good balance between outer-loop loss and inner-loop loss which were completed in separate stages in the Meta-Baseline method; the end-to-end sequential choice of the optimisation strategy is justified by looking at validation loss over time which degrades in the previous baseline.
- The experiments are extensive and establish the superior performance of the method with statistical significance. It's shown that Boost-MT also achieves better transferability between different dataset domains.
- I also really appreciate the fact that empirical claims within the paper were justified by direct results and the language of said claims were accordingly adjusted.

**Weaknesses:**

- The paper's writing is convoluted at times and requires more digging than usual to learn some details of the method. I strongly recommend that the authors simplify the notation and include Algorithm 1 in the main body of the paper. The key aspect of the method where the loss function during the inner loop which updates the feature extractor is updated using a loss in each task that draws both the feature extractor in the outer loop and the very last update is a key point that may be missed by most readers.
- Meta-Baseline is the primary baseline the paper compares to, especially in cross-dataset evaluation. The addition of more baselines can further strengthen the empirical evaluation.

**Questions:**

- Why does it seem to have the 5-15 range be a sweet spot for several inner loop iterations per outer loop? Does the author have any additional studies regarding the patterns observed in this direction?
- What is the role of mu term in Equation 6? Furthermore, is there any empirical evaluation of the magnitude of the inner and outer losses when gradient updates are applied? Does one term seem to dominate?
- Overall, I believe the proposed method is interesting, simple yet empirically powerful, and would be happy to improve my current rating once the authors address the questions and concerns noted, especially around improving the paper's notation and writing.

---

### Official Review · Reviewer_gX3A · 2023-11-01

**Soundness:** 3 good
**Presentation:** 2 fair
**Contribution:** 2 fair
**Rating:** 5
**Confidence:** 4

**Summary:**

The authors propose a new training method for meta-learning, which improves over Meta-Baseline. Instead of two-stage training of pre-training and meta-training used in Meta-baseline, they aim to perform end-to-end training where they simultaneously utilize base class examples and novel examples at meta-training stage. They validate their method on miniImagenet and tieredImagenet benchmarks with ablation experiments.

**Strengths:**

- The motivation for the proposed method is clear and straightforward, with formulation that is simple and easy to understand.

- Their methodology is technically plausible, while solving a practical problem with possibility for application to wide range of few-shot learning tasks.

- Their approach shows competitive performance on some well-known few-shot learning and cross-domain few-shot learning benchmarks.

- They validate their method under various hyperparameter settings with ablation experiments to back up their claims.

**Weaknesses:**

- Compared to the previous Meta-baseline, the only difference of the proposed method seems to be in the composition of the mini-batch used in the outer loop computation. What is the difference of the proposed method compared to Meta-Baseline with occasional base class gradient updates/corrections?

- Pretrained weights of backbone feature extractors are very easy to come by, especially for ResNets. Are there ablation experiments where classification model is initialized with pretrained weights, and meta-trained using the proposed method?

- Improvements of using base class information is marginal, where improvement over baseline algorithm Meta-Baseline seems to be within the margin of error.

- Since more general few-shot learning benchmark such as Meta-Dataset [a] is available, additional validations will further help to back up the effectiveness of the proposed method.

  [a] Meta-Dataset: A Dataset of Datasets for Learning to Learn from Few Examples, Triantafillou et al., ICLR 2020

- Some of the sentences were hard to understand with grammar issues and ambiguity. I encourage the authors to fully proofread their manuscript before any form of publication.

- Minor point: Since Figure 1 shows the motivation of the proposed method, it would be helpful to include the validation accuracy plots for the proposed method (Figure 3(b)) along with other methods.

**Questions:**

Please refer to the weaknesses above. I appreciate some insights provided in the paper, but there are some limitations in the method's novelty and experimental validations.

---

### Official Review · Reviewer_jif1 · 2023-11-10

**Soundness:** 3 good
**Presentation:** 2 fair
**Contribution:** 2 fair
**Rating:** 3
**Confidence:** 4

**Summary:**

The main contributions of this paper are threefold:
- The authors focus on analyzing the popular meta-training procedures with two consecutive learning phases, i.e., the conventional pretraining and the episodic training.
- Based on the inefficient learning behavior during the two-phase training, which implies the probable conflicts between the objectives of the two phases, the authors combine them in a single training phase by designing a bi-level optimization process comprising two objectives for the conventional classification training and the few-shot training.
- The authors have confirmed that the proposed training called Boost-MT shows performance gains over the prior methods.

**Strengths:**

**Strength 1:** To the best of my knowledge, the efforts to scrutinize the learning behaviors of the conventional two-phase training for meta-learning have remained unexplored. Although the two-phase training is widely used as a routine process, not much is known about the reason why the process shows performance gains beyond the single-phase episodic training. Also, we do not know much about the optimality of the two-phase process in training deep models with strong generalization. This work has found that the two-phase training shows somewhat conflicting behaviors between phases (as shown in Fig. 1), which makes re-design the learning procedure of few-shot learners. That is the most interesting and valuable contribution of this paper.

**Strength 2:** The proposed single-phase training process is fully model-agnostic, and we can even say that it is algorithm-agnostic. Therefore, I expect the proposed method can be widely applied to the prior meta-learners and other possible future learners.

**Weaknesses:**

**Weakness 1:** The main concern is that the paper does not explicitly verify the synergetic effects between the inner (i.e., training of classifiers via a conventional full-class classification task) and outer (i.e., episodic training with a few samples) optimizations. In my opinion, the most important and fundamental question is the synergetic effect between conventional training and episodic training. Some authors have tried to focus on the differences between conventionally-trained and meta-trained feature extractors (Goldblum et al., 2020), but not many facts are unexplored yet. Therefore, I expect this work to unveil the role of the inner and outer optimization of the proposed method to fully understand the effects of the full-class classification objective and the few-shot training objective, respectively.

**Weakness 2:** More experiments for validating the applicability of the proposed method would be required. I believe that one of the biggest advantages of this method is its wide applicability. Because this work touches on the training procedure, which is algorithm-agnostic, potential gains from applying the training procedure to a wide range of meta-learners would clearly elucidate the novelty of this method. However, only a naive version of 'Boost MT', which applies the training procedure to a simple ProtoNet style model, is experimented with. Will the method yield consistent gains when applied to other early meta-learners, including TPN (Liu et al., 2019) and TADAM (Oreshkin et al., 2018), or relatively recent approaches, including CAN (Hou et al., 2019) and STANet (Lai et al., 2023)? The first group of prior works is advanced metric-based meta-learners, and the second group is based on attention or transformer to boost the few-shot learning performance, which is now positioned as the state-of-the-art.

**Weakness 3:** Further ablation studies that verify the algorithmic procedure in the proposed training should be provided. To the best of my understanding, I cannot find any ablation studies that answer the question of why we have to devise training procedures with the inner and outer structures. Also, what is the reason behind the losses for the inner and outer optimizations? What happens when we drop one of the inner and outer optimizations? In fact, these questions are closely related to 'Weakness 1' for verifying the synergetic advantages of the proposed training procedures.

There are some minor errors:
- In Abstract, it would be better to say "Also," rather than "And" at the beginning of a sentence.
- In Introduction, please add a citation of 'Meta-Baseline', which is directly analyzed in the motivating experiment shown in Fig. 1.

M. Goldblum et al. "Unraveling Meta-Learning: Understanding Feature Representations for Few-Shot Tasks," ICML 2020.
Y. Liu et al., "LEARNING TO PROPAGATE LABELS: TRANSDUCTIVE PROPAGATION NETWORK FOR FEW-SHOT LEARNING," ICLR 2019
R. Hou et al., "Cross Attention Network for Few-shot Classification," NeurIPS 2019.
J. Lai et al., "SpatialFormer: Semantic and Target Aware Attentions for Few-Shot Learning," AAAI 2023.

**Questions:**

**Q1:** Would you provide an in-depth explanation of the role of the inner and outer optimizations of the proposed method?

**Q2:** What are the synergetic effects between the inner and outer optimizations? Are there any empirical results or conceptual insights that explain the particular training procedure?

**Q3:** When combined with further complicated meta-learners (rather than a simple ProtoNet style version in the reported results), will Boost MT yield consistent gains? Specifically, the prior works aforementioned in Weakness 2 can be additional baselines, or other prior algorithms that are already compared in Tables 1 and 2 can be a baseline to apply Boost MT. When specifying the question, what is the accuracy of 'Boost MT + Method' beyond 'Method'?